# Up-Front ASCT Overcomes the Survival Benefit Provided by HDAC-Based Induction Regimens in Mantle Cell Lymphoma: Data from a Real-Life and Long-Term Cohort

**DOI:** 10.3390/cancers15194759

**Published:** 2023-09-28

**Authors:** Luís Alberto de Pádua Covas Lage, Marcela do Vale Elias, Cadiele Oliana Reichert, Hebert Fabrício Culler, Fábio Alessandro de Freitas, Renata de Oliveira Costa, Vanderson Rocha, Sheila Aparecida Coelho da Siqueira, Juliana Pereira

**Affiliations:** 1Department of Hematology, Hemotherapy & Cell Therapy, Faculty of Medicine, University of São Paulo (USP), São Paulo 05508-080, SP, Brazil; marcela.vale@hc.fm.usp.br (M.d.V.E.);; 2Laboratory of Medical Investigation in Pathogenesis and Directed Therapy in Onco-Immuno-Hematology (LIM-31), Faculty of Medicine, University of São Paulo (USP), São Paulo 05508-080, SP, Brazil; 3Department of Hematology and Hemotherapy, Faculty of Medical Sciences of Santos (FCMS), Santos 01238-010, SP, Brazil; 4Hospital Alemão Osvaldo Cruz (HAOC), São Paulo 01323-020, SP, Brazil; 5Fundação Pró-Sangue, Blood Bank of São Paulo, São Paulo 05403-000, SP, Brazil; 6Department of Hematology, Churchill Hospital, Oxford University, Oxford OX3 7LE, UK; 7Department of Pathology, University of São Paulo (USP), São Paulo 05508-080, SP, Brazil

**Keywords:** mantle cell lymphoma (MCL), immunochemotherapy, high-doses cytarabine-based induction regimens (HDAC), autologous stem cell transplantation (ASCT), treatment, prognosis

## Abstract

**Simple Summary:**

This study aimed to assess clinical outcomes, determine survival predictors, and compare responses between different primary therapeutic modalities in a large real-world cohort of patients with mantle cell lymphoma (MCL), with a focus on assessing the impact of intensified immunochemotherapy regimens based on high doses of cytarabine (HDAC) on outcomes in ASCT-eligible patients. A total of 165 Brazilian patients with biopsy-proven MCL were included from 2010 to 2022. After a long follow-up, our results demonstrated that patients treated with (R)-HDAC-based regimens had higher ORR (85.9% vs. 65.7%, *p* = 0.007) compared to those treated with (R)-CHOP, as well as lower rates of early relapses (61.9% vs. 80.4%, *p* = 0.043) and lower mortality (43.9% vs. 68.6%, *p* = 0.004). However, enhanced induction regimens employing (R)-HDAC were not associated with a real overall survival benefit in MCL patients undergoing ASCT (2-year OS: 88.7% for (R)-HDAC plus ASCT vs. 78.8% for (R)-CHOP plus ASCT, *p* = 0.289). Additionally, up-front ASCT was independently associated with improvement in OS (*p* < 0.001), EFS (*p* = 0.005), and POD-24 (*p* < 0.001) in MCL. In conclusion, in the largest real-world Latin American study involving MCL patients, we were able to ratify the benefit of up-front ASCT in young and physically fit patients regardless of the intensity of the induction immunochemotherapy regimen used. Although HDAC-based induction regimens were not associated with improved survival in ASCT-eligible patients, it was associated with higher ORR and lower rates of early relapses in the whole cohort. These findings can decisively impact the therapeutic management of MCL patients in different clinical settings.

**Abstract:**

Background: Mantle cell lymphoma (MCL) is a rare malignancy with heterogeneous behavior. Despite the therapeutic advances recently achieved, MCL remains incurable. Currently, the standard of care for young and fit patients involves induction immunochemotherapy followed by up-front autologous stem cell transplantation (ASCT). However, the role of more intensive induction regimens, such as those based on high doses of cytarabine (HDAC), remains controversial in the management of ASCT-eligible patients. Methods: This retrospective, observational, and single-center study involved 165 MCL patients treated at the largest oncology center in Latin America from 2010 to 2022. We aimed to assess outcomes, determine survival predictors, and compare responses between different primary therapeutic strategies, with a focus on assessing the impact of HDAC-based regimens on outcomes in ASCT-eligible patients. Results: The median age at diagnosis was 65 years (38–89 years), and 73.9% were male. More than 90% of the cases had a classic nodal form (cnMCL), 76.4% had BM infiltration, and 56.4% presented splenomegaly. Bulky ≥ 7 cm, B-symptoms, ECOG ≥ 2, and advanced-stage III/IV were observed in 32.7%, 64.8%, 32.1%, and 95.8%, respectively. Sixty-four percent of patients were categorized as having high-risk MIPI. With a median follow-up of 71.1 months, the estimated 2-year OS and EFS were 64.1% and 31.8%, respectively. Patients treated with (R)-HDAC-based regimens had a higher ORR (85.9% vs. 65.7%, *p* = 0.007) compared to those receiving (R)-CHOP, as well as lower POD-24 rates (61.9% vs. 80.4%, *p* = 0.043) and lower mortality (43.9% vs. 68.6%, *p* = 0.004). However, intensified induction regimens with (R)-HDAC were not associated with a real OS benefit in MCL patients undergoing up-front consolidation with ASCT (2-year OS: 88.7% vs. 78.8%, *p* = 0.289). Up-front ASCT was independently associated with increased OS (*p* < 0.001), EFS (*p* = 0.005), and lower POD-24 rates (*p* < 0.001) in MCL. Additionally, CNS infiltration, TLS, hypoalbuminemia, and the absence of remission after induction were predictors of poor OS. Conclusions: In the largest Latin American cohort of MCL patients, we confirmed the OS benefit promoted by up-front consolidation with ASCT in young and fit patients, regardless of the intensity of the immunochemotherapy regimen used in the pre-ASCT induction. Although HDAC-based regimens were not associated with an unequivocal increase in OS for ASCT-eligible patients, it was associated with higher ORR and lower rates of early relapses for the whole cohort.

## 1. Introduction

Mantle cell lymphoma (MCL) is a malignancy characterized by clonal proliferation of small to medium-sized atypical B-lymphocytes with heterogeneous biological behavior and recurrently associated with the genetic rearrangement t(11;14) (q13;q32) and cyclin-D1 (CCND1) expression [1,2]. It is the third most common subtype of non-Hodgkin’s lymphoma (NHL) in Western countries, accounting for 5–7% of all lymphomas [1,2]. North American and European statistics indicate an incidence of 1 new case per 200,000 inhabitants per year [3]. However, epidemiological data from developing countries is scarce, and its real incidence is virtually unknown in these regions.

MCL preferentially affects elderlies, with a median age at diagnosis around 60–70 years and a clear predominance in males (70% of cases) [1,3,4]. Clinically, MCL is characterized by disseminated lymphadenopathies, accompanied or not by B-symptoms. Few patients are diagnosed in early stages (Ann Arbor I/II) or with localized extranodal disease, while more than 80% of cases present advanced stages III/IV, and more than 50% exhibit bone marrow (BM) involvement [1,3,4]. MCL patients usually have good *performance status* (ECOG 0–1), only 25% present bulky disease ≥ 7 cm, and high lactate dehydrogenase (LDH) levels occur in less than 50% of cases [3,4]. Other typical presentations include bowel involvement, characterized as multiple intestinal polyps (“*lymphomatoid polyposis*”) with predisposition to ileocecal intussusception, as well as a truly leukemic form, observed in up to 10–15% of cases [1].

Currently, two distinct clinical–biological subtypes of MCL are recognized, including the classic nodal mantle cell lymphoma (cnMCL) and the non-nodal leukemic variant (lMCL) [1,2]. The cnMCL corresponds to 80–90% of all cases, presents aggressive behavior, is characterized by involvement of lymph nodes and extra-lymphoid tissues, and may also frequently affect the BM and, less commonly, the peripheral blood. Biologically, cnMCL demonstrates overexpression of the SOX-11 protein and negativity for CD200, as well as an unmutated status of the *IgVH* gene (un*IgVH*). This subtype also presents high genomic instability, characterized by epigenetic dysregulation and abnormalities involving the *ATM*, *CDKN2A,* and *TP53 genes*. On the other hand, the leukemic variant (lMCL) corresponds to only 10–20% of MCL cases, has an indolent behavior, manifests as giant splenomegaly, absence of adenopathy, BM involvement, and presence of lymphocytosis, which may be exuberant (>50–100 × 10^9^/L). Characteristically, low biological aggressiveness is observed in this subtype, which shares many clinical and biological findings with chronic lymphocytic leukemia (CLL). Additionally, positivity for CD200 and absence of SOX-11 expression by neoplastic cells are observed, as well as a low grade of genomic instability and often mutated-*IgVH* status (m*IgVH*) [1,3,5].

Classically, the MCL’s cell of origin (COO) is an unmutated-*IgVH* pre-germinal center B-cell. Its primary clonogenic event is t(11;14) (q13;q32), which is present in up to 85% of cases. Through this chromossomal translocation, the juxtaposition of the *CCND1/IgVH* genes occurs, leading to cyclin-D1 overexpression. Cyclin-D1 is an oncoprotein with cell cycle “accelerating” properties. Its overexpression promotes the activation of the CDK4/6 complex, with consequent phosphorylation and inactivation of the retinoblastoma (Rb) protein, leading to the transition from the G1 phase to the S phase (G1/S) and consequently activating tumor cell proliferation (“*pro-proliferative effect*”). Secondary genetic events are associated with MCL progression and transformation to blastoid subtypes, including loss of tumor suppressor genes such as *ATM*, *CDKN2A,* and *TP53* and gain of the oncogenes *MYC*, *SYK,* and *BCL2*, justifying its biological heterogeneity [4,6,7,8]. Furthermore, the complex interactions between neoplastic cells and different elements from the immune tumor microenvironment (TME) promote activation of intracellular signaling pathways, such as BCR (*B-cell receptor*) and NF-kB (*nuclear factor-kappa B*), contributing to the progression and dissemination of MCL [7,8].

MCL exhibits a markedly heterogeneous prognosis, highly influenced by clinical–biological factors. In this sense, the first tool with the capacity to predict outcomes was the *Mantle Cell International Prognostic Index* (MIPI), developed in 2008 by Hoster E. et al. [9]. This prognostic score includes the independent variables age, LDH, ECOG, and total WBC count, stratifying MCL patients into low-risk (<5.7 points), intermediate-risk (5.7–6.1), and high-risk (≥6.2) categories, presenting medians of overall survival (OS) not reached, of 51 months and 29 months, respectively [9]. The same study group demonstrated that the nuclear protein related to cell proliferation, Ki-67, accessed by immunohistochemistry (IHC), proved to be a biological factor with strong and independent prognostic value in MCL, particularly being associated with poor outcomes when expressed by more than 30% of tumor cells. This observation led to the incorporation of Ki-67 into the classical MIPI variables and the subsequent development of a new prognostic score known as biological-MIPI (B-MIPI, MIPI plus Ki-67). B-MIPI promoted a clear OS distinction between low-/intermediate-risk categories and the high-risk group [10]. Additionally, different groups described the adverse prognostic impact of other variables in MCL, including blastoid cytology, complex karyotype, and *TP53* mutation [10,11,12].

Recently, different strategies have been used for the first-line treatment of MCL, varying from exclusive observation (“*watchful & waiting approach*”) to intensified immunochemotherapy regimens followed by up-front consolidation with autologous stem cell transplantation (ASCT). Young patients (aged < 60–65 years) and “*fit*” (absence of severe comorbidities and with ECOG < 2) are usually considered eligible for ASCT and may experience induction therapy based on rituximab plus high-doses of cytarabine (HDAC), such as R-DHAP [rituximab, dexamethasone, HDAC, cisplatin], R-DHAOX [rituximab, dexamethasone, HDAC, oxaliplatin], R-CHOP/R-DHAP, and R-HyperCVAD [R-CHOP alternated with rituximab, methotrexate plus HDAC], followed by consolidation with high-dose therapy (HDT) and ASCT, as well as rituximab maintenance for at least 2 years. MCL patients considered ineligible for ASCT (elderlies and/or “*unfit*”) may receive a variety of non-enhanced immunochemotherapy regimens (not based on HDAC), such as R-CHOP [rituximab, cyclophosphamide, doxorubicin, and prednisone], BR [bendamustine, rituximab], VR-CAP [bortezomib, rituximab, cyclophosphamide, adriamycin, and prednisone], or R2 [rituximab plus lenalidomide], followed by maintenance with rituximab [3,4,13,14]. Bruton tyrosine kinase inhibitors (BTKi) (ibrutinib, acalabrutinib, and zanubrutinib) are approved as second-line therapy for MCL in the context of relapsed/refractory (R/R) disease. However, recent trials were conducted in order to promote the incorporation of these agents as first-line therapy in combination with immunochemotherapy [15]. Particular attention should be given to patients harboring the *TP53* mutation, who characteristically have dismal outcomes and high rates of resistance to traditional immunochemotherapy regimens [3,4,15].

Although great advances have been obtained in MCL management in the last two decades, the disease still remains incurable, and a high proportion of cases do not experience sustained remissions. Therefore, knowledge of its clinical–biological characteristics, identification of prognostic factors, and a better understanding of its first-line therapeutic possibilities and the responses obtained by MCL patients when submitted to these different approaches are of great utility to direct personalized and risk-adapted strategies. Based on this premise, we carried out an observational study in the largest cohort of Brazilian patients with MCL described so far, followed by a long-term follow-up, aiming to identify independent prognostic factors associated with survival or early progression. Additionally, we intend to assess responses and outcomes promoted by different first-line modalities, focusing on defining the real impact provided by HDAC-based induction regimens in ASCT-eligible patients.

## 2. Methods

### 2.1. Study Design and Ethical Issues

This is an observational, retrospective, and single-center study carried out at the Instituto do Câncer do Estado de São Paulo (ICESP)/Hospital das Clínicas, Faculty of Medicine, University of São Paulo (HC-FMUSP), Brazil. It was approved by the local Ethics Committee in 2023 (CAAE number: 65603822.3.0000.0068), with the application of the Free and Informed Consent Form (FICF) to all living participants included in the study and with their respective consents. The waiver of the FICF application was obtained for dead patients and with non-recoverable contact. All clinical-demographic, laboratory, pathological, imaging, and therapeutic data were obtained from institutional electronic medical records.

### 2.2. Patients and Eligibility Criteria

Considering all 1138 patients diagnosed with NHL and registered in the *ICESP/HC-FMUSP Non-Hodgkin’s Lymphoma Database*, 180 patients diagnosed with MCL were initially selected. Of these, 165 (91.7%) were included in this study, all of them with a biopsy-proven diagnosis of MCL. Eligibility criteria included age ≥ 18 years old, patients diagnosed and treated at ICESP/HC-FMUSP from January 2010 to December 2022, and a confirmed or strongly presumptive diagnosis of MCL. The diagnostic criteria applied for MCL included the following: demonstration of a mature B-cell lymphoproliferative neoplasm with one or more of the following requirements: 1. positivity for cyclin-D1 by IHC; 2. expression of CD5 and SOX-11 by IHC for cyclin-D1 negative cases; 3. presence of t(11;14) (q13;q32) by conventional karyotyping (G banding) in BM or peripheral blood samples, 4. presence of t(11;14) (q13;q32) by fluorescent in situ hybridization (FISH) in formalin-fixed and paraffin-embedded (FFPE) tumor samples, BM, or peripheral blood, 5. presence of the *CCND1/IgVH* transcript accessed by RT-PCR in FFPE tumor, BM, or peripheral blood samples, 6. immunophenotyping by flow cytometry of peripheral blood or BM showing a mature and monoclonal B-cell lymphoid population with coexpression of the markers CD5^+^/CD200^−^ and Matutes score ≤ 3 points. Exclusion criteria involved all CD5^+^ and cyclin-D1-negative mature B-cell lymphoproliferative disorders with a Matutes Score equal to 4 or 5 points and/or those with high clinical-laboratory, phenotypic, and histopathological suspicion of chronic lymphocytic leukemia (CLL), splenic marginal zone lymphoma (SMZL), or lymphoplasmacytic lymphoma/Waldesntröm’s macroglobulinemia (LPL/WM), as well as cases of monoclonal B-cell lymphocytosis (BML) with an MCL-like phenotype. After applying the eligibility criteria, 15 cases were excluded (8.3%), including 11 cases with insufficient clinical-laboratory information and 4 cases with a distinct diagnosis of MCL (atypical CLL [N = 2], SMZL [N = 1], and WM [N = 1]).

Clinical-epidemiological, laboratory, and histopathological data accessed at diagnosis included age, gender, ethnicity, performance status by *Eastern Cooperative Oncology Group* (ECOG), *Ann Arbor/Cotswolds* clinical stage, clinical–biological MCL subtype (cnMCL vs. lMCL), presence of peripheral blood involvement (“leukemic phase”), BM infiltration, central nervous system (CNS), gastrointestinal tract, Waldeyer’s ring, orbits, skin and subcutaneous tissue involvement, splenomegaly (≥13 cm), bulky disease (≥7 cm), B-symptoms, tumor lysis syndrome (TLS), presence of ≥2 comorbidities, and *Mantle Cell International Prognostic Index* (MIPI) score. Laboratory variables related to blood cell count (hemoglobin, WBC, neutrophils, lymphocytes, monocytes, neutrophil/lymphocyte ratio [N/L], lymphocyte/monocyte ratio [L/M], and total platelet count), LDH, β2-microglobulin, albumin, globulins, and viral serological status for HIV and hepatitis B and C were also accessed. Histopathological variables included the diagnostic biopsy site, percentage of cases with blastoid morphology, frequency of cases with Ki-67 expression > 30%, positivity for CD5, CD23, CD200, and cyclin-D1.

Date of diagnosis, complete response (CR), disease relapse and progression, first and last cycle of chemotherapy and radiotherapy, date and cause of death, and date of last follow-up were obtained to assess the overall response rate (ORR), complete response (CR), partial response (PR), overall survival (OS), event-free survival (EFS), and progression of disease within the first 24 months of starting upfront therapy (POD-24). At diagnosis, all MCL patients underwent a complete blood count, comprehensive biochemical tests, including renal and hepatic function and tests for TLS, as well as LDH and β2-microglobulin measurements, and serology for HIV and hepatitis B and C. All patients eligible for anthracycline-based induction regimens underwent transthoracic ecocardiography to estimate left ventricule function (LVF).

Staging was carried out with contrast-enhanced computed tomography (CT) scans of the neck, chest, abdomen, and pelvis, or preferably by CT with positron emission with 18-fluor-deoxyglucose (18-FDG-PET-CT) and unilateral BM biopsy with IHC. Selected cases underwent complementary tests, including immunophenotyping by flow cytometry, karyotyping by G-banding, FISH for t(11;14) (q13;q32), and/or RT-PCR for the *CCND1/IgVH* fusion. Patients with localized stage (I/II) or with gastrointestinal symptoms were submitted to evaluation of the gastrointestinal tract with upper digestive endoscopy and colonoscopy with multiple biopsies, in the case of the former, to exclude occult microscopic involvement of the gastrointestinal tract and subsequent confirmation of early-stage disease. Patients with blastoid morphology and/or Ki-67 > 30% underwent cerebrospinal fluid (CSF) puncture with chemocytological analysis as part of the staging procedures.

### 2.3. Up-Front Therapeutic Management, Response Assessment and Follow-Up

Up-front treatment of newly diagnosed MCL patients followed the current institutional protocols at the time of diagnosis. In brief, patients with asymptomatic lMCL (without B-symptoms, massive splenomegaly, or severe cytopenias) were managed with a “*watchful & waiting*” approach, while those with symptomatic disease experienced monotherapy with rituximab 375 mg/sqm I.V. weekly for 4 to 6 consecutive doses or immunochemotherapy based on the (R)-CHOP regimen [rituximab 375 mg/sqm I.V. on D1, cyclophosphamide 750 mg/sqm I.V. on D1, doxorubicin 50 mg/sqm I.V. on D1, vincristine 1.4 mg/sqm [max 2 mg] I.V. on D1, and prednisone 100 mg/day P.O. on D1 to D5] for 6 to 8 courses with an interval of 21 days between cycles.

Patients with cnMCL presenting early-stage (I/II) disease received involved-field radiotherapy (IF-RT) with 30–36 Gy (15 to 18 fractions of 200 cGy). Individuals with advanced-stage cnMCL (III/IV) underwent different immunochemotherapy protocols depending on their eligibility for consolidation with ASCT. Young (aged < 65 years) and “fit” patients (ECOG ≤ 2 and without severe comorbidities) diagnosed before 2015 experienced induction immunochemotherapy with 6 to 8 cycles of the (R)-HyperCVAD regimen [course A: rituximab 375 mg/sqm I.V. on D1, cyclophosphamide 300 mg/sqm I.V. on D1-D3, vincristine 1.4 mg/sqm (max 2 mg) I.V. on D4 and D11, doxorubicin 60 mg/sqm I.V. on D4 and dexamethasone 40 mg/day P.O. on D1–D4 and D11–D14; course B: rituximab 375 mg/sqm I.V. on D1, methotrexate 1000 mg/sqm I.V. on D1, cytarabine 3000 mg/sqm I.V. on D2-D3, followed by urinary alkalization with sodium bicarbonate and administration of folinic acid 18 h after the end of methotrexate infusion for 3–5 days. Young and “fit” cnMCL patients diagnosed after 2015 experienced induction therapy with 6 cycles of the (R)-CHOP/(R)-DHAP hybrid regimen with a 21-day interval between cycles [(R)-CHOP, as described previously, alternated with (R)-DHAP: rituximab 375 mg/sqm I.V. on D1, cisplatin 100 mg/sqm I.V. on D1, cytarabine 2000 mg/sqm every 12 h on D2, dexamethasone 40 mg/day P.O. on D1–D4]. Patients with CR/PR at the end of induction regimens experienced consolidation with HDT followed by autologous stem cell rescue (ASCT).

cnMCL patients considered ineligible for ASCT (elderly and/or “unfit”) were submitted to first-line therapy with 6 to 8 cycles of (R)-CHOP-21 or anthracycline-based regimens with dose reductions, such as (R)-miniCHOP [(R)-CHOP with dose reduction of doxorubicin to 25 mg/sqm I.V. on D1] and elderly (R)-miniCHOP [rituximab 375 mg/sqm I.V. on D1, cyclophosphamide 400 mg/sqm I.V. on D1, vincristine 1.0 mg/sqm (max 2 mg) I.V. on D1, and prednisone 40 mg/sqm P.O. on D1–D5]. Few patients were conducted in clinical research protocols, such as VR-CAP (bortezomib, rituximab, cyclophosphamide, adriamycin, and prednisone), BRi (bendamustine, rituximab, and ibrutinib), or BRa (bendamustine, rituximab, and acalabrutinib), or received oral chemotherapy with palliative intent, such as chlorambucil 10 mg/sqm P.O. on D1–D6 for 8 to 12 courses. No case in our cohort experienced maintenance immunotherapy with rituximab because this therapeutic modality is not provided for MCL patients treated in the Brazilian public health system.

MCL patients with blastoid morphology and/or Ki67 > 30% received CNS chemoprophylaxis with intrathecal injections of methotrexate (12 mg) and dexamethasone (2 mg) administered in the first 4 cycles of the induction regimens. Those cases with proven CNS infiltration experienced intrathecal injections of methotrexate 12 mg, cytarabine 100 mg, and dexamethasone 2 mg twice a week until CSF cytology was negative, followed by weekly administrations for 4 consecutive weeks, and finally monthly administrations for 12 months. Patients with high tumor burden, ECOG > 2, or a high probability of developing TLS underwent cytoreduction chemotherapy with one cycle of the CVP regimen [cyclophosphamide 300 mg/sqm I.V. on D1, vincristine 1 mg/sqm (max 2.0 mg) I.V. on D1, and prednisone 40 mg/sqm P.O. on D1-D7] before starting the induction phase. Infectious prophylaxis with trimethoprim-sulfamethoxazole and/or acyclovir, as well as the use of growth factors (filgastrim), were applied in selected cases in accordance with institutional practice.

Response evaluation was performed after the fourth cycle and at the final induction course, as well as thirty days after ASCT in cases eligible for ASCT, according to the criteria proposed by Cheson BD et al. in 2014 [16]. All patients with BM involvement at diagnosis and who were in first CR by imaging studies at the end of the induction regimen underwent a new BM biopsy. Clinical follow-up was done every 3 months for the first two years after CR, every 6 months for the third and fourth years, and annually after the fifth year.

### 2.4. Histopathology, Immunohistochemistry and Immunophenotyping by Flow Cytometry

All cases were submitted to centralized histopathological review by two experts in hematopathology and categorized as MCL according to the *Classification of Hematopoietic and Lymphoid Tissue Neoplasms* proposed by the *World Health Organization* (WHO) in 2016 [1]. Tumor biopsies obtained at diagnosis were formalin-fixed and paraffin-embedded (FFPE). FFPE sections of 2 µm were displayed on silanized slides and stained with hematoxylin and eosin (H&E) for initial analysis. Immunohistochemistry was carried out with the monoclonal antibodies CD45 (Dako, 2B11 + PD7/26, 1/200), Ki-67 (Dako, J55, 1/1600), pan-B CD20 (Dako, L26, 1/1000), pan -T CD3 (Dako, F7.2.38, 1/500), CD10 (Novocastra, S6C6, 1/2000), CD5 (Invitrogen, 53–7.3, 1/200), CD23 (Sigma–Aldrich, 1B12, 1/500), cyclin-D1 (Invitrogen, SP4, 1/1000), and SOX-11 (Cell-Marque, MRQ-58, 1/200).

Additionally, patients presenting lymphocytosis (≥5 × 10^9^/L) and/or BM infiltration underwent immunophenotyping by flow cytometry on a *BD FACSCalibur^®^* 4-color cytometer (San Jose, CA, USA) with a panel composed by the monoclonal antibodies CD20 (ExBio, Vestec, the Czech Republic, 2H7, PerCp, 5 µL), CD79b (ExBio, HM79, FITC, 5 µL), CD5 (ExBio, CRIS1, PE, 5 µL), CD3 (ExBio, UCHT1, FITC, 5 µL), CD200 (BD Pharmigen, 552475, PE, 5 µL), CD19 (IOTest, PC5.5, PerCp, 5 µL), FMC7 (IOTest, A07791, FITC, 5 µL), CD23 (ExBio, EBVCS-5, PE, 5 µL), cykappa (ExBio, A8B5, PE, 5 µL), and cylambda (ExBio, 4C2, PE, 5 µL). Figure 1 demonstrates typical cases of pleomorphic and blastoid MCL diagnosed by combining PB/BM cytomorphology and immunophenotyping by flow cytometry.

### 2.5. Statistical Analysis

Data were shown in accordance with the variables evaluated. Categorical variables were presented in absolute (N) and relative (%) values. Numerical variables were presented as measures of central tendency (median), dispersion (min-max range), and position. The median follow-up time was calculated using the reverse Kaplan–Meier method (reverse KM). Analysis of overall survival (OS) and event-free survival (EFS) was performed using the Kaplan–Meier (KM) method, and the Log-Rank test was used for comparison between treatment groups. The Chi-square test was applied to assess statistically significant differences in clinical-laboratory characteristics and responses between different treatment arms. This analysis was plotted in a heatmap to highlight the main differences found in the form of aggregation by microgroups. OS was considered from the date of diagnosis to death; EFS from the date of diagnosis to the date of progression; death from any cause; or last follow-up. Progression of disease within 24 months (POD-24) was considered the percentage of cases that experienced disease progression or relapse within 24 months of up-front treatment. The data were censored at the last follow-up.

Analysis to determine predictors for outcomes, including those related to OS, EFS, and POD-24, was performed using Cox’s semiparametric univariate method. Multivariate analysis using a multiple-step Cox’s regression model was conducted to determine independent prognostic variables. All variables with a *p*-value ≤ 0.10 identified in univariate analysis were included in the final model for multivariate analysis. The results were presented in Hazard Ratio (HR), 95% Confidence Interval (95% CI), and forest graphics. All analyses were performed using the statistical software SPSS version 28.0 for Windows, and a *p*-value ≤ 0.05 was considered statistically significant.

## 3. Results

### 3.1. Clinical, Demographic, Laboratorial and Histopathological Features

Clinical-demographic, laboratory, and pathological findings of the 165 MCL patients were displayed in Table 1, Table 2 and Table 3. The median age was 65 years (range: 38–89 years), with 66.7% (110/165) of the elderly (aged ≥ 60 years old). One hundred twenty-two patients (73.9%) were male, and 72.1% (119/165) were Caucasian. Regarding the clinical presentation, 90.9% (150/165) had cnMCL, and only 9.1% (15/165) presented lMCL. Peripheral blood involvement (leukemic phase) was observed in 47.3% (78/165) and BM infiltration in 76.4% (120/157). CNS, gastrointestinal tract, Waldeyer’s ring, orbits, and skin/subcutaneous tissue involvement occurred in 6.1% (10/165), 22.4% (37/165), 20% (33/165), 3.6% (6/165), and 4.8% (8/165), respectively.

Splenomegaly (≥13 cm) was observed in 56.4% (93/165) of cases, while bulky disease (≥7 cm) and B-symptoms occurred in 32.7% (54/165) and 64.8% (107/165), respectively. ECOG ≥ 2 was observed in 32.1% (53/165); however, the majority of patients (95.8%—158/165) had advanced-stage (Ann Arbor III/IV) at diagnosis. The presence of ≥2 clinical comorbidities was verified in 49.1% (81/165), while only 6.1% (10/165) developed spontaneous or treatment-triggered TLS. The median MIPI prognostic score was 6.4 points (range: 4.1–13.7 points), with 13.3% (22/165) categorized as low-risk (<5.7 points), 22.4% (37/165) categorized as intermediate-risk (5.7–6.1 points), and 64.2% (106/165) stratified as high-risk (≥6.2 points).

The medians of hemoglobin, total white blood cell (WBC) count, neutrophils, lymphocytes, monocytes, and platelets were 114 g/L, 8.72 × 10^9^/L, 3.75 × 10^9^/L, 2.51 × 10^9^/L, 0.60 × 10^9^/L, and 160 × 10^9^/L, respectively. The medians of neutrophil/lymphocyte (N/L) and lymphocyte/monocyte (L/M) ratios were 1.34 and 4.26, respectively. The medians of LDH, patient LDH/control LDH ratio, β2-microglobulin, albumin, and total globulins were 357 U/L, 1.07, 3.80 mg/dL, 4.0 g/dL, and 2.5 g/dL, respectively. Anemia occurred in 57% (94/165), leukocytosis in 37.6% (62/165), neutrophilia in 17% (28/165), lymphocytosis in 43% (71/165), and monocytosis in 25.5% (42/165), as summarized in Table 2. More than half of the whole cohort had high LDH (55.8%—92/165), 67.1% (88/131) had elevated levels of β2-microglobulin, hypoalbuminemia was registered in 72.9% (113/155), and hypogammaglobulinemia occurred in 2.6% (4/150) of cases. Additionally, 5.5% (9/154) had a monoclonal component in serum protein electrophoresis, and positivity for HIV, HbsAg, and anti-HCV occurred in 0.6% (1/158), 0% (0/155), and 1.3% (2/155) of cases, respectively.

In 50.9% (84/165) of cases, the diagnostic biopsy was obtained from nodal sites, in 23.6% (39/165) from extra lymphoid tissues, and in 25.5% (42/165) from BM. In the histopathological and immunohistochemical study, blastoid morphology was observed in 17.6% (29/165) of the tissue specimens, a high proliferative index (Ki-67 ≥ 30%) in 56.5% (65/115), and expression of the markers CD5, CD23, and cyclin-D1 in 91.5% (150/164), 6.8% (9/132), and 95.1% (157/165). Negativity for the CD200 antigen was observed in 93.2% (69/74) of the cases, assessed in samples of peripheral blood and/or bone BM using immunophenotyping by flow cytometry. The histopathological data are compiled in Table 3.

### 3.2. Up-Front Therapeutic Modalities

Among the 165 MCL patients included in the study, 7.9% (13/165) did not experience any antineoplastic therapy, being that 4.9% (8/165) presented early death during staging procedures and before starting up-front therapy, and 3.0% (5/165) had asymptomatic lMCL and were managed with a “*watchful & waiting*” approach. Among the 152 (92.1%) effectively treated cases, 46.0% (70/152) received an induction regimen with (R)-CHOP, 43.4% (66/152) were treated with HDAC-based immunochemotherapy, and of the latter, 37.8% (25/66) received the (R)-HyperCVAD regimen, and 62.2% (41/66) were treated with the hybrid protocol (R)-CHOP/(R)-DHAP. Just over 5% of cases (8/152) were treated in the context of clinical research trials with the VR-CAP, BRi, and BRa regimens; palliative therapy based on chlorambucil monotherapy was used in 4.6% (7/152); and only one patient (0.7%) received isolated IF-RT. Cytoreduction based on the CVP regimen was used in 41.4% (63/152) of cases, while CNS prophylaxis with the administration of intrathecal chemotherapy was applied in 22.3% (34/152) of cases. Only 21.2% (32/151) received rituximab in addition to induction polychemotherapy regimens, and 7.3% (11/151) underwent IF-RT as an adjunct to primary therapy. Consolidation with HDT/ASCT was used in 32.8% (50/152) of treated cases; of these, 41/50 (82%) experienced ASCT in the first CR/PR, while 9/50 (18%) experienced this modality in the second CR/PR. More than half of the treated patients (51.9%—79/152) required ≥ 2 lines of therapy; however, allogeneic stem cell transplantation (alloSCT) was only applied in 3.0% (5/165) of the cases, all of them with relapsed disease after ASCT. Table 4 summarizes the therapeutic management of the whole cohort.

### 3.3. Responses and Outcomes

The overall response rate (ORR) for the whole cohort was 64.7% (95% CI: 57.1–71.7%), with CR achieved in 41.5% (95% CI: 34.1–49.1%) and PR in 23.2% (95% CI: 17.2–30.0%). Primary refractory disease occurred in 25.5% (95% CI: 19.2–32.4%) of cases. POD-24 was observed in 46.7% (95% CI: 39.1–54.3%). The overall mortality rate (OMR) during all follow-up was 58.2% (95% CI: 50.6–65.5%), with 30.2% (29/96) of deaths due to disease progression, 26% (25/96) were due to infectious complications, 19.7% (19/96) occurred due to overlap between disease progression and active infection, 4.2% (4/96) were attributed to cardiovascular and/or thromboembolic complications, 3.1% (3/96) were associated with second neoplasms, and in 16.7% (16/96) the cause of death could not be defined, usually in cases with death registered outside the oncological service of origin.

The median follow-up was 71.1 months (95% CI: 61.8–80.4 months). The median overall survival (OS) was 39.7 months (95% CI: 27.9–51.6 months), with an estimated 2-year OS of 64.1% (95% CI: 56.2–71.9%) and 36.6% (95% CI: 28.1–45.0%) in 5 years (Figure 2A). Similarly, the median event-free survival (EFS) was 13.8 months (95% CI: 9.3–18.3 months), and the estimated 2-year and 5-year EFS were 31.8% (95% CI: 23.0–40.6%) and 8.4% (95% CI: 1.9–14.8%), respectively (Figure 2B). Additionally, the median disease-free survival (DFS) was 20.5 months (95% CI: 16.0–25.0 months), with an estimated 2-year and 5-year DFS of 39.1% (95% CI: 29.3–48.9%) and 21.2% (95% CI: 12.5–29.8%), respectively (Figure 2C).

### 3.4. Responses and Outcomes According to Up-Front Therapies

Among the 152 patients with MCL who experienced any type of anti-cancer therapy, 136 (89.5%) received induction immunochemotherapy with (R)-CHOP-like (N = 70) or with regimens based on (R)-HDAC (N = 66). Figure 3 presents a heatmap summarizing the main associations observed between clinical–biological data and first-line treatment applied to MCL patients. In brief, patients treated with (R)-CHOP-like were older (aged ≥ 60 years in 81.4% [57/70] vs. 43.9% [29/66], *p* < 0.001), had a higher percentage of the lMCL form (11.4% [8/70] vs. 1.5% [1/55], *p* = 0.02), and presented a higher number of comorbidities (≥2 comorbidities in 58.6% [41/70] vs. 33.3% [22/66], *p* = 0.003) than cases treated with (R)-HDAC. Additionally, patients treated with (R)-CHOP-like had a higher percentage of high-risk MIPI (77.1% [54/70] vs. 48.5% [32/66], *p* = 0.002), as well as lower remission rates (65.7% [44/67] vs. 85.9% [55/64], *p* = 0.007), higher rates of POD-24 (80.4% [45/56] vs. 61.9% [26/42], *p* = 0.043), and higher mortality (68.6% [48/70] vs. 43.9% [29/66], *p* = 0.004) when compared to those receiving (R)-HDAC induction.

Concerning outcomes, MCL patients treated with (R)-CHOP induction had a median OS of 26.5 months (95% CI: 18.5–34.4 months), while those receiving (R)-HDAC showed a median OS of 55.4 months (95% CI: 26.1–84.7 months). The estimated 2-year OS was 58% (95% CI: 45.8–70.1%) for the (R)-CHOP group and 73% (95% CI: 61.6–84.3%) for the (R)-HDAC group, *p* < 0.001, as summarized in Figure 4A. The median EFS was 13.4 months (95% CI: 8.3–18.4 months) for cases treated with (R)-CHOP and 18.8 months (95% CI: 12.1–25.5 months) for those treated with (R)-HDAC. The estimated 2-year EFS was 25.7% (95% CI: 13.5–37.8%) for the (R)-CHOP arm versus 40% (95% CI: 23.7–56.2%) for the (R)-HDAC arm, *p* = 0.07 (Figure 4B).

Patients undergoing up-front ASCT had a median OS not reached, whereas patients not eligible for ASCT had a median OS of only 26 months (95% CI: 16.5–35.5%). The estimated 2-year OS was 86.4% (95% CI: 75.2–97.6%) for patients consolidated with up-front ASCT versus only 55.7% (95% CI: 46.5–65%) for those who did not experience this therapeutic modality, *p* < 0.001 (Figure 4C). Similarly, the median EFS was 32.3 months (95% CI: 10.1–54.5 months) for ASCT-consolidated MCL cases and 12.1 months (95% CI: 9–15.2 months) for those ineligible for ASCT. We observed an estimated 2-year EFS of 60% (95% CI: 35.3–85%) for cases submitted to up-front ASCT, versus only 28.2% (95% CI: 19–37.4%) for patients not eligible for this therapy, *p* = 0.004, as summarized in Figure 4D.

In a pooled analysis involving the induction therapy regimen and up-front consolidation with ASCT for chemosensitive patients, we observed median OS not reached for the (R)-HDAC plus up-front ASCT group, 79 months (95% CI: 0–166.6 months) for the (R)-CHOP plus up-front ASCT group, 36.7 months (95% CI: 11.4–61.9 months) for the group receiving only (R)-HDAC-based induction without consolidation based on ASCT, and 25.9 months (95% CI: 20.1–31.7 months) for those group treated exclusively with (R)-CHOP. Similarly, the estimated 2-year OS were 88.7% (95% CI: 76.5–100%), 78.8% (95% CI: 52.5–100%), 59.8% (95% CI: 42.7–76.8%), and 53.2% (95% CI: 39.6–66.7%) for the groups (R)-HDAC plus up-front ASCT, (R)-CHOP plus up-front ASCT, (R)-HDAC without ASCT, and isolated (R)-CHOP, respectively, *p* < 0.001. Figure 5 demonstrates the OS curves for MCL patients included in the study according to primary treatment strategy, including up-front chemoimmunotherapy induction regimens and consolidation.

### 3.5. Prognostic Factors

In univariate analysis, the variables associated with decreased OS were age ≥ 60 years [HR: 1.95, 95% CI: 1.22–3.13, *p* = 0.005], peripheral blood involvement [HR: 1.47, 95% CI: 0.98–2.20, *p* = 0.061], CNS infiltration [HR: 3.33, 95% CI: 1.72–6.44, *p* < 0.001], ECOG ≥ 2 [HR: 3.03, 95% CI: 1.73–5.30, *p* < 0.001], ≥2 comorbidities [HR: 1.43, 95% CI: 0.95–2.14, *p* = 0.081], TLS [HR: 5.15, 95% CI: 2.39–11.09, *p* < 0.001], anemia (Hemoglobin < 120 g/L) [HR: 1.55, 95% CI: 1.02–2.34, *p* = 0.037], leukocytosis [HR: 2.68, 95% CI: 1.29–5.56, *p* = 0.008], lymphocytosis [HR: 2.32, 95% CI: 1.13–4.77, *p* = 0.022], hypoalbuminemia [HR: 2.28, 95% CI: 1.50–3.47, *p* < 0.001], MIPI score high-risk category [HR: 2.89, 95% CI: 1.25–6.64, *p* = 0.012], absence of remission [HR: 4.38, 95% CI: 2.82–6.79, *p* < 0.001], POD-24 [HR: 3.02, 95% CI: 1.86–4.89, *p* < 0.001], upfront regimens not-based on HDAC [HR: 2.20, 95% CI: 1.38–3.51, *p* < 0.001], and omission of up-front ASCT [HR: 5.59, 95% CI: 2.80–11.17, *p* < 0.001].

Predictors for decreased EFS were cnMCL subtype [HR: 2.10, 95% CI: 1.04–4.22, *p* = 0.037], CNS involvement [HR: 2.07, 95% CI: 1.05–4.06, *p* = 0.034], orbit infiltration [HR: 2.84, 95% CI: 0.89–9.06, *p* = 0.076], ECOG ≥ 2 [HR: 2.09, 95% CI: 1.19–3.67, *p* = 0.009], TLS [HR: 3.79, 95% CI: 1.77–8.13, *p* < 0.001], anemia [HR: 1.76, 95% CI: 1.15–2.69, *p* = 0.008], high LDH levels [HR: 1.60, 95% CI: 1.06–2.42, *p* = 0.024], high-risk category by MIPI score [HR: 2.11, 95% CI: 0.91–4.96, *p* = 0.079], no remission after induction therapy [HR: 2.65, 95% CI: 1.70–4.14, *p* < 0.001], and omission of up-front consolidation with ASCT [HR: 2.56, 95% CI: 1.32–4.96, *p* = 0.005]. HDAC-based induction regimens were associated with increased EFS in univariate analysis [HR: 0.66, 95% CI: 0.41–1.06, *p* = 0.092].

We identified as predictors for POD-24: age ≥ 60 years [HR: 2.34, 95% CI: 1.35–4.06, *p* = 0.002], CNS involvement [HR: 3.86, 95% CI: 1.91–7.80, *p* < 0.001], TLS [HR: 3.93, 95% CI: 1.86–8.29, *p* < 0.001], anemia [HR: 1.65, 95% CI: 1.03–2.63, *p* = 0.034], leukocytosis [HR: 2.29, 95% CI: 1.01–5.18, *p* = 0.046], lymphocytosis [HR: 2.25, 95% CI: 0.95–5.32, *p* = 0.064], high-risk MIPI score [HR: 3.09, 95% CI: 1.14–7.72, *p* = 0.016], omission of HDAC in the induction regimen [HR: 2.21, 95% CI: 1.35–3.62, *p* = 0.001], and omission of up-front ASCT [HR: 4.91, 95% CI: 2.35–10.23, *p* < 0.001]. Figure 6A–C summarizes the main results obtained by univariate analysis to identify prognostic factors associated with OS (Figure 6A), EFS (Figure 6B), and POD-24 (Figure 6C).

In multivariate analysis, the independent prognostic factors associated with decreased OS in our MCL cohort were CNS involvement [HR: 3.12, 95% CI: 1.45–6.73, *p* = 0.004], TLS [HR: 2.79, 95% CI: 1.13–6.86, *p* = 0.026], albumin < 3.5 g/dL [HR: 2.69, 95% CI: 1.61–4.50, *p* < 0.001], failure to achieve remission after induction therapy [HR: 3.71, 95% CI: 2.07–6.67, *p* < 0.001], and omission of up-front ASCT [HR: 4.59, 95% CI: 2.23–9.42, *p* < 0.001]. Similarly, TLS [HR: 4.29, 95% CI: 1.81–10.18, *p* < 0.001], absence of remission after induction therapy [HR: 10.97, 95% CI: 5.53–21.76, *p* < 0.001], and absence of up-front consolidation with ASCT [HR: 2.29, 95% CI: 1.12–4.68, *p* = 0.023] were independently associated with decreased EFS. Finally, the independent adverse factors associated with the occurrence of early progression (POD-24) were anemia [HR: 2.32, 95% CI: 1.37–3.94, *p* = 0.002], failure to achieve remission after induction therapy [HR: 3.16, 95% CI: 1.79–5.58, *p* < 0.001], and omission of up-front ASCT [HR: 4.80, 95% CI: 2.20–10.46, *p* < 0.001] (Figure 7).

## 4. Discussion

Here we reported the main clinical-laboratory characteristics, response rates, and outcomes according to first-line therapeutic modalities in the largest Latin American cohort of MCL patients treated in a real-life setting. We also assessed the main predictors associated with clinical outcomes, including OS, EFS, and early relapses (POD-24), in this population. The identification of such prognostic factors is fundamental for better understanding MCL behavior in different geographic areas since most of the cohorts currently reported are from North America and Europe, and Latin American data are virtually unknown. Additionally, the determination of prognostic factors and response predictors allows the establishment of risk-adapted therapeutic strategies for MCL patients.

Currently, the therapeutic management of young (aged ≤ 65 years) and physically fit MCL patients is centered on induction immunochemotherapy regimens containing high-dose cytarabine (HDAC), followed by up-front consolidation with ASCT in chemosensitive cases [3,4]. The importance of adding HDAC to MCL induction regimens has been emphasized by several study groups over the last decade and has become the current standard of care for young and fit MCL patients. Some authors even advocate that patients undergoing HDAC-based regimens, such as R-HyperCVAD, do not require consolidation with ASCT if they reach first CR after the induction course [17]. Therefore, a phase II trial involving 97 newly diagnosed MCL patients conducted by researchers at the *MD Anderson Cancer Center* (MDACC) reported ORR and CR of 97% and 87%, respectively, after induction treatment with the R-HyperCVAD regimen without up-front ASCT consolidation [18]. Two other smaller studies, from the Italian group and the *Southwestern Oncology Group* (SWOG), also reported high ORRs of 83% and 86%, respectively, after induction with R-HyperCVAD. However, in both trials, haematological toxicities were not uncommon, leading to early therapy discontinuation in a considerable portion of patients [19,20].

Despite the benefits promoted by induction regimens based on HDAC in MCL, up-front consolidation with ASCT promotes increased overall survival and is being incorporated into the primary therapeutic strategy of young and fit patients with MCL by many international research groups [21,22,23,24,25]. In this sense, trials conducted by the Nordic group involving more than 160 MCL patients treated with the R-maxi-CHOP/R-HDAC regimen followed by up-front consolidation with ASCT reported an ORR of 96%, with a CR of only 54% for patients not undergoing ASCT and 89% for those who have experienced ASCT. Overall, 145/160 patients underwent ASCT, with a median OS and PFS of 12.7 years and 8.5 years, respectively. Just less than 10% of patients developed secondary malignancies, and although remissions were durable, 50% of cases relapsed after 12 years, reinforcing the incurable character of this disorder, even when patients are submitted to intensified therapeutic modalities [22,23,24].

Aiming to address the real impact of HDAC on the chemotherapeutic induction of MCL, French researchers conducted a phase II study with the hybrid regimen R-CHOP/R-DHAP followed by ASCT in 60 patients newly diagnosed with MCL. This regimen was intended to minimize the influence of other drugs, such as high-dose methotrexate (used in the R-HyperCVAD regimen) and high-dose cyclophosphamide (used in the R-maxi-CHOP regimen), which appear to play a less important role in controlling MCL and may add considerable toxicity. In that study, with a median follow-up of 67 months, an ORR of 94% and an estimated 5-year OS of 75% were obtained [26]. Based on these preliminary results, the *European MCL Network* conducted a phase III study aimed at confirming the superiority of the R-CHOP/R-DHAP regimen followed by ASCT over the R-CHOP regimen followed by ASCT [25]. This prospective trial conducted by Hermine et al. included 466 MCL patients with a median follow-up of 6.1 years. Despite the HDAC-based regimen promoting a longer time to treatment failure (9.1 years vs. 4.3 years, HR: 0.56, *p* = 0.038), the ORR (98% vs. 97%) and CR (63% vs. 61%) did not differ between both treatment groups. Additionally, the OS was not significantly different between both arms (HR: 0.78, 95% CI: 0.57–1.07, *p* = 0.12), and a significant number of secondary malignancies were observed in the HDAC group, including 2.4% of secondary leukemias and 4.3% of solid tumors [25].

In fact, numerous methodological differences exist between the study conducted by Hermine et al. and our study, and these ones prevent a precise comparison between the results obtained in both studies. Therefore, Hermine et al. conducted a multicenter and prospective clinical trial, randomized and matched 1:1, initially involving 497 MCL patients. That cases were highly selected (ECOG ≤ 2, age ≤ 65 years) to receive up-front therapy based on R-CHOP plus ASCT versus R-CHOP/R-DHAP followed by ASCT. Of these, effectively 466 cases were analyzed and followed for a long time (median follow-up: 6.1 years). At the same time, our study is retrospective and unicentric; it involved 165 unselected MCL patients and was conducted in a real-life setting. Our median follow-up was also long (5.9 years). Only a quarter of our population (24.8%—41/165) underwent up-front consolidation with ASCT; however, in this particular subgroup, we observed, in agreement with what was demonstrated by Hermine et al., that patients with MCL undergoing induction with R-HDAC-based regimens did not show an increased OS compared to patients treated with induction based on R-CHOP regimens.

In our real-world study, considering the whole cohort, MCL patients treated with HDAC-based induction regimens had higher ORR (85.9% vs. 65.7%, *p* = 0.007), lower rates of early relapses/POD-24 (61.9% vs. 80.4%, *p* = 0.043), and lower overall mortality (43.9% vs. 68.6%, *p* = 0.004) than those treated with R-CHOP. Furthermore, in univariate analysis, HDAC-based regimens provided benefits in OS (2-year OS: 73% vs. 58%, *p* < 0.001) and EFS (2-year EFS: 40% vs. 25.7%, *p* = 0.07). However, this survival benefit was highly influenced by the fact that most patients undergoing (R)-HDAC-based induction experienced up-front consolidation with ASCT. Therefore, in multivariate analysis, induction regimens based on HDAC did not have an impact as an independent prognostic variable associated with increased overall survival, but up-front consolidation with ASCT did. Also, in multivariate analysis, the omission of ASCT in the primary regimen of MCL patients was associated with a four times higher risk of mortality (HR: 4.59, 95% CI: 2.23–9.42, *p* < 0.001), a two times higher risk of developing an event (HR: 2.29, 95% CI: 1.12–4.68, *p* = 0.023), and an almost five times higher risk for early progression/relapse (HR: 4.80, 95% CI: 2.20–10.46, *p* < 0.001).

Such data confirm the real survival benefit promoted by up-front consolidation with ASCT in young and physically fit patients with MCL considered eligible for this therapeutic strategy, regardless of whether the induction regimen is based on the use of HDAC. Pooled analysis, including the 4 main primary therapy arms applied in our cohort (Figure 5), demonstrated that patients treated with R-HDAC plus ASCT had similar OS compared to those treated with R-CHOP plus ASCT (2-year OS: 88.7% for R-HDAC plus ASCT vs. 77.8% for R-CHOP followed by ASCT, *p* = 0.289). As previously mentioned, these data are in agreement with those presented by Hermine et al. (2016) and also with those observed in a retrospective review of the *National Comprehensive Cancer Network* (NCCN) database, which demonstrated that R-CHOP alone promotes lower PFS compared to immunochemotherapy followed by ASCT; however, the outcomes do not differ between R-CHOP followed by ASCT and more intensive induction regimens, such as R-HyperCVAD followed by ASCT [27]. In brief, the data presented in our study indicate that young and fit MCL patients should preferably be induced with HDAC-based immunochemotherapy followed by up-front consolidation with ASCT. However, when it is not possible to offer HDAC, R-CHOP-21 followed by ASCT can be a safe and effective first-line modality. It promotes a similar OS benefit since up-front consolidation with ASCT leads to a substantial increase in OS for MCL despite whether the induction regimen contains or does not contain HDAC. Recently, Albanyan et al. (2023) and Ng Zi Yun et al. (2019) also demonstrated that the intensity of the induction regimen does not impact post-transplant outcomes in MCL patients undergoing ASCT in their first complete remission [28,29].

Although the benefit of up-front ASCT seems unequivocal for the therapeutic management of MCL patients considered eligible for this therapeutic modality, its toxicities and mortality are not negligible. In this sense, based on the favorable results observed with the use of Bruton tyrosine kinase (BTK) inhibitors in R/R MCL patients [30,31,32,33], new studies have recently been conducted to incorporate these agents into primary MCL therapeutic regimens. Therefore, there is a possibility that the use of ibrutinib or acalabrutinib associated with immunochemotherapy may overcome the need for consolidation with up-front ASCT in young and fit patients with MCL. The TRIANGLE study (NCT02858258) may clarify this issue. This is an ongoing phase III trial involving patients with MCL eligible for ASCT and prospectively randomized into three arms: R-CHOP/R-DHAP plus ASCT, R-CHOP/R-DHAP plus ibrutinib followed by ASCT and consolidation with ibrutinib, or R-CHOP/R-DHAP plus ibrutinib followed by consolidation with ibrutinib for two years. Recently, the benefit of adding ibrutinib to immunochemotherapy (BR) regimens was demonstrated in the first-line setting for ASCT-ineligible MCL patients with a significant advantage in PFS (80.6 months for the BR plus ibrutinib group vs. 52.9 months for the BR group plus placebo, *p* = 0.01) [15].

In our study, ORR (64.7%) and RC (41.5%), as well as OS (estimated 2-year OS 64.5%) and EFS (estimated 2-year EFS 31.8%), were similar to those reported by international groups in the pre-rituximab and pre-ASCT eras but lower than those reported in more recent studies. These data are summarized in Table 5, which characterizes responses and clinical outcomes in different MCL cohorts over the last two decades. However, our study has numerous peculiarities. Among them, it is a real-life study, including a population with poor clinical and laboratory characteristics. Here, nearly 50% of patients had two or more comorbidities, and 64.2% were categorized as high-risk according to the MIPI score. Additionally, the primary therapeutic strategies applied to our cohort were very heterogeneous: only 21.2% of the cases had access to immunotherapy in an induction regimen; no patient experienced maintenance therapy with rituximab; and only about a third of the cohort experienced consolidation with HDT/ASCT. All these factors can explain the responses and outcomes presented in our study. It should also be noted that prospective studies involving MCL patients in the post-rituximab and post-ASCT eras included a highly selected and homogeneous population, usually with good performance status, the absence of serious comorbidities, and frequently low tumor volumes.

However, when we performed a subgroup analysis in our population, those young and fit patients (aged ≤ 65 years, ECOG ≤ 2, and with low comorbidities index) treated with intensive immunochemotherapy followed by up-front consolidation with ASCT showed responses and outcomes similar to those observed in MCL cases included in recent large collaborative studies. In this sense, considering our population, patients treated with (R)-HDAC followed by up-front ASCT had a median OS not reached and an estimated 2-year OS of 88.7%, and patients treated with (R)-CHOP plus ASCT had a median OS of 79 months and an estimated 2-year OS of 78.8%.

Here, we also identified that CNS infiltration, tumor lysis syndrome (TLS), hypoalbuminemia, failure to obtain CR after first-line therapy, and omission of up-front ASCT were adverse and independent prognostic factors associated with decreased OS in Brazilian patients with MCL. Several collaborative international groups listed adverse prognostic factors associated with shortened OS in MCL, many of which were similar to those found in our cohort (Table 6). The identification of these prognostic factors is of great importance to identify patients with a greater probability of treatment failure or early progression and who consequently may benefit from more intensified therapeutic strategies or the up-front use of new agents, such as BTK inhibitors or consolidation with chimeric antigen receptor T-cell (CAR-T) therapies.

Among the limitations presented by our study, we highlight those intrinsic to retrospective studies, as well as the fact that we had a cohort that was submitted to heterogeneous therapeutic modalities, with limited access to immunotherapy, both in induction and maintenance regimens, in addition to the absence of genetic-molecular characterization of the included cases. Currently, we know that maintenance therapy with rituximab at a dose of 375 mg/sqm I.V. applied every 2 months for at least 2 years improves OS and PFS, both in ASCT-eligible and non-eligible MCL patients [21,41]. However, maintenance immunotherapy with rituximab is not funded by the Brazilian public health system (SUS—*Sistema Único de Saúde*) for patients with MCL, which may explain, at least partially, the lower survival outcomes found in our cohort when compared to the North American and European series. Despite some limitations, this is the largest real-world Latin American study ever reported, which confirms the importance of up-front consolidation with ASCT in MCL management, regardless of the induction immunochemotherapy regimen used. Knowledge of such data is essential to improving standards of care in MCL, emphasizing the need to consolidate eligible patients with HDT/ASCT, particularly in constrained-resource settings where access to immunotherapy and new drugs is not universal.

## 5. Conclusions

In the largest real-life Latin American cohort including MCL patients, up-front consolidation with ASCT was associated with increased OS regardless of the intensity of the induction immunochemotherapy regimen applied. Here, we ratify the overall survival benefit promoted by up-front ASCT in MCL, which independently overcomes the advantage provided by HDAC in the pre-transplant induction. Although HDAC-based regimens were not associated with overall survival-independent benefits in ASCT-eligible patients, their use was related to higher ORR and lower rates of disease progression within 24 months from initial therapy. Additionally, CNS infiltration, TLS, hypoalbuminemia, absence of remission after first-line treatment, and omission of up-front ASCT predicted decreased OS in Brazilian patients with MCL.

## Figures and Tables

**Figure 1 cancers-15-04759-f001:**
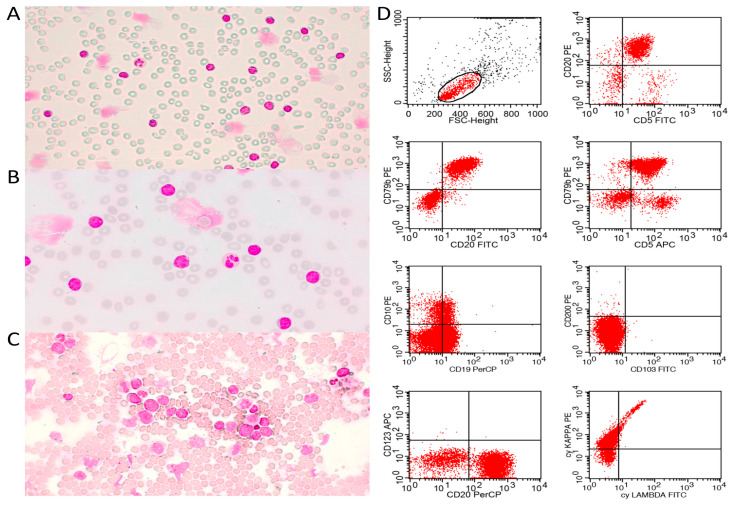
Typical cases of MCL. Peripheral blood smear demonstrating proliferation of small to medium-sized atypical lymphoid cells, pleomorphic, with presence of smudge cells ((**A**)—optical microscopy, 10× magnification; (**B**)—optical microscopy, 40× magnification, Leishman staining). Bone marrow smear with proliferation of medium- to large-sized atypical lymphocytes, with an immature aspect, in a case of blastoid MCL ((**C**)—optical microscopy, 40× magnification, Leishman staining). (**D**)—Immunophenotyping by flow cytometry demonstrated proliferation of small cells with low internal complexity and a mature B-lymphoid phenotype: CD20^++bright^, CD79b^++bright^, CD5^+^, CD19^+dim^, CD200^−^, cykappa^+^. Negativity for cylambda^−^, CD10^−^, CD103^−^ and CD123^−^ markers.

**Figure 2 cancers-15-04759-f002:**
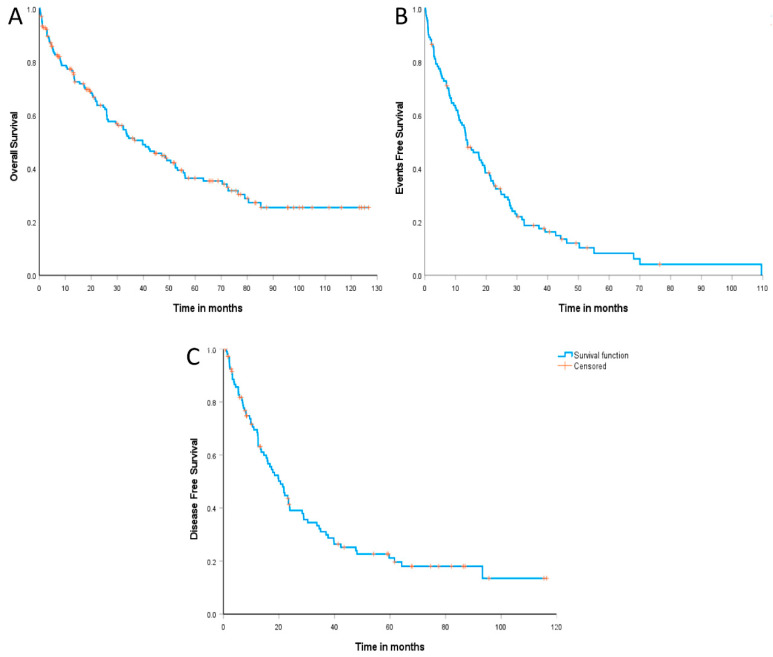
OS (**A**), EFS (**B**), and DFS (**C**) of 165-Brazilian patients with MCL included in the study.

**Figure 3 cancers-15-04759-f003:**
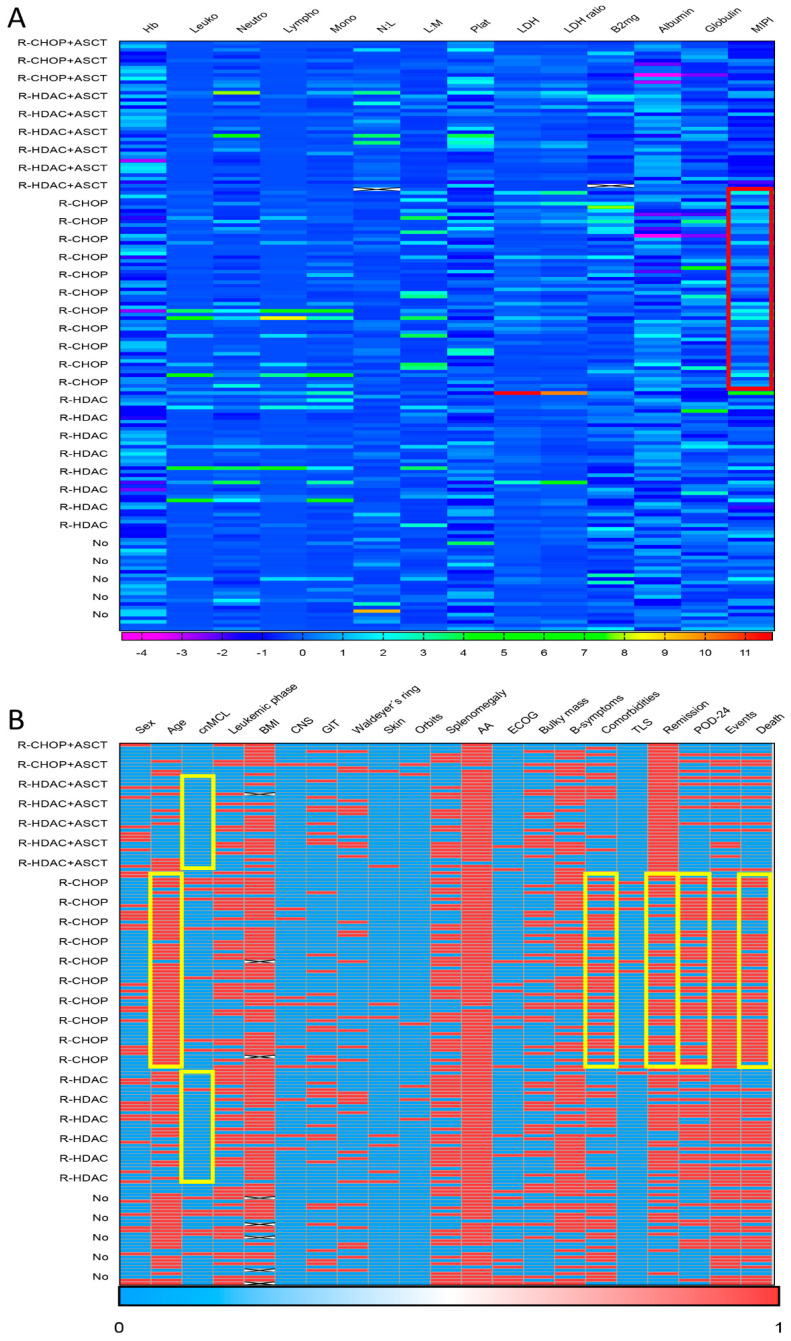
Heatmap demonstrating statistically significant associations between clinical–biological and evolutionary characteristics of the 165 MCL patients and up-front therapeutic modalities. In (**A**), the red box demonstrates high aggregation of patients categorized as high-risk MIPI among cases treated with (R)-CHOP (*p* = 0.002). In (**B**), the yellow boxes show a high concentration of elderly patients (*p* < 0.001), with a higher number of comorbidities (*p* = 0.003), higher rates of early relapses (POD-24) (*p* = 0.043) and mortality (*p* = 0.004), as well as lower rates of remission (*p* = 0.007) in the (R)-CHOP group, in addition to a higher frequency of the cnMCL variant in patients treated with (R)-HDAC-based regimens (*p* = 0.02). Legend: Hb: hemoglobin; Leuko: leukocyte count; Neutro: neutrophils; Lympho: lymphocytes; Mono: monocytes; N/L: neutrophil/lymphocyte ratio; L/M: lymphocyte/monocyte ratio; Plat: platelets count; LDH: lactic dehydrogenase; LDH ratio: LDH patient/LDH control (upper value of normality) ratio; B2mg: B2-microglobulin; MIPI: *Mantle Cell Lymphoma International Prognostic Index;* cnMCL: classic nodal mantle cell lymphoma; BMI: bone marrow involvement; CNS: central nervous system; GIT: gastrointestinal tract; AA: Ann Arbor staging; ECOG: *Eastern Cooperative Oncology Group*; TLS: tumor lysis syndrome; POD-24: progression of disease within 24 months; ASCT: autologous stem cell transplantation; 0: absence; 1: presence.

**Figure 4 cancers-15-04759-f004:**
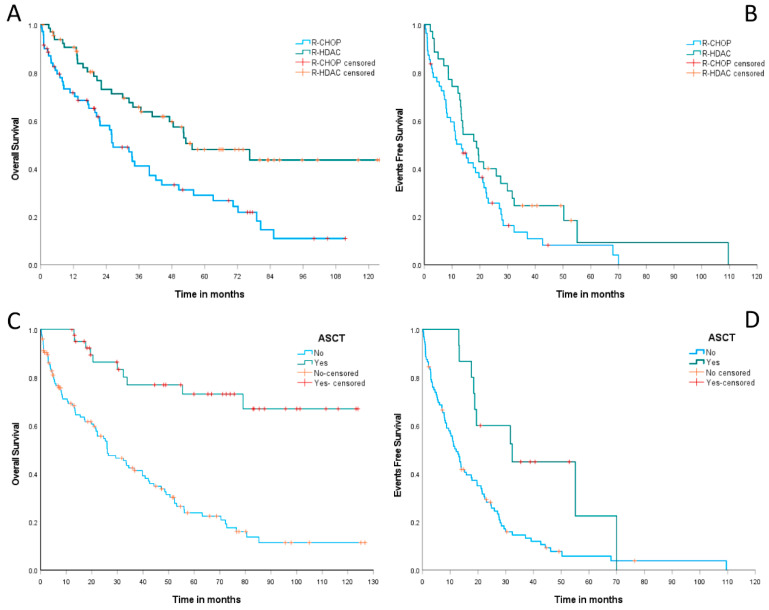
OS (**A**)—*p* < 0.001 and EFS (**B**)—*p* = 0.07 curves according to up-front induction regimens (R-CHOP vs. R-HADC regimens). OS (**C**)—*p* < 0.001 and EFS (**D**)—*p* = 0.004 curves according to up-front consolidation strategies (ASCT vs. no-ASCT).

**Figure 5 cancers-15-04759-f005:**
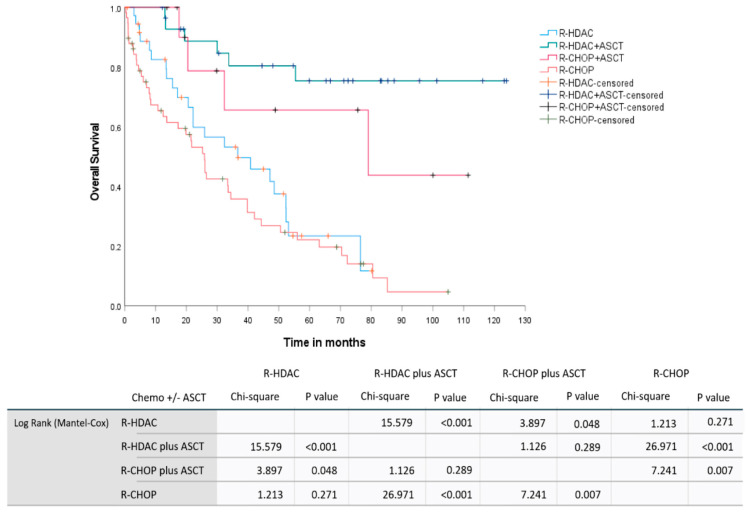
OS curves according to the complete up-front therapeutic modalities, involving induction regimens and consolidation with ASCT in first CR/PR. Green: (R)-HDAC induction followed by up-front ASCT; red: (R)-CHOP induction followed by up-front ASCT; blue: (R)-HDAC induction not followed by ASCT; orange: isolated (R)-CHOP induction. The Log-Rank test for comparison between pairs demonstrates statistically significant differences between the groups treated with chemo plus ASCT and those treated with chemo alone (R-HDAC plus ASCT/R-HDAC *p* < 0.001; R-HDAC plus ASCT/R-CHOP *p* < 0.001; R-CHOP plus ASCT/R-HDAC *p* = 0.048; R-CHOP plus ASCT/R-CHOP *p* = 0.007), but no statistically significant difference was demonstrated between both groups treated with chemo plus ASCT (R-HADC plus ASCT/R-CHOP plus ASCT *p* = 0.289) and neither between both groups treated with isolated induction immunochemotherapy (R-HDAC/R-CHOP *p* = 0.271).

**Figure 6 cancers-15-04759-f006:**
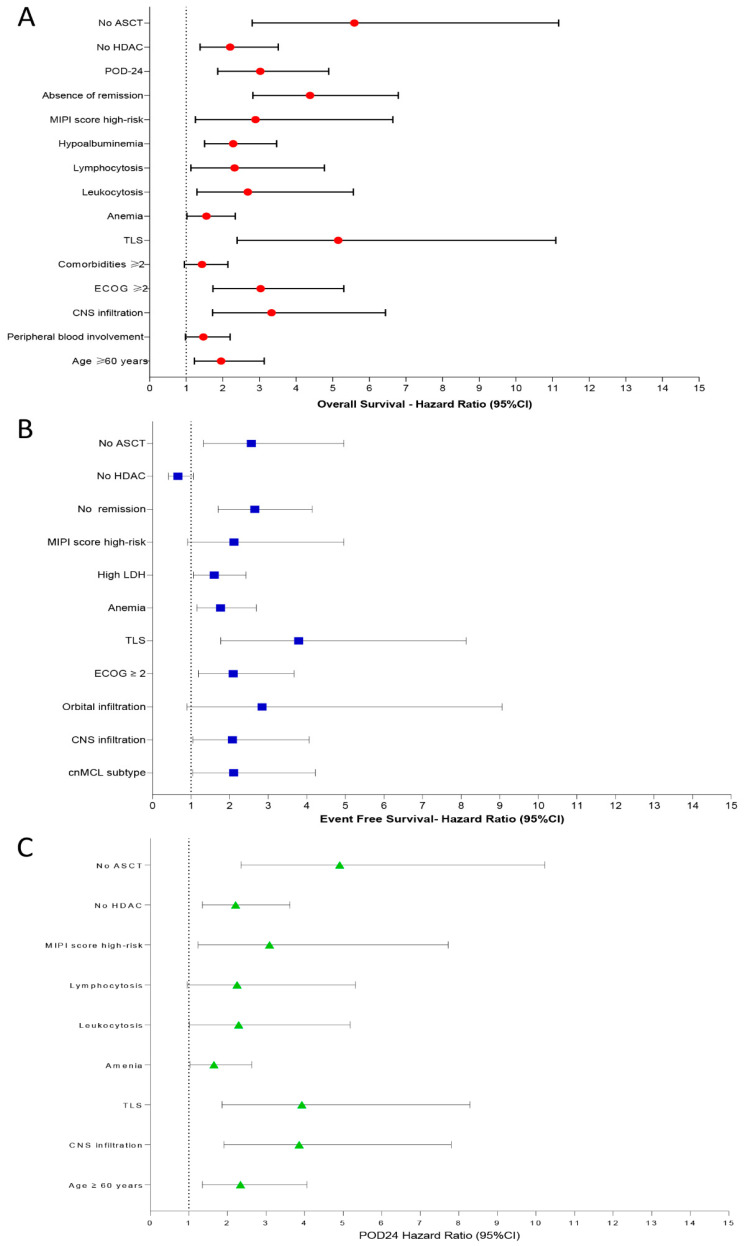
Predictive factors for OS (**A**), EFS (**B**), and POD-24 (**C**) in 165 MCL patients identified in univariate analysis. Legend: ASCT: autologous stem cell transplantation; HDAC: high-dose cytarabine-based induction chemotherapy; POD-24: progression of disease within 24 months; MIPI: *Mantle Cell Lymphoma International Prognostic Index*; TLS: tumor lysis syndrome; ECOG: *Eastern Cooperative Oncology Group*; CNS: central nervous system; LDH: lactic dehydrogenase; cnMCL: classic nodal mantle cell lymphoma variant.

**Figure 7 cancers-15-04759-f007:**
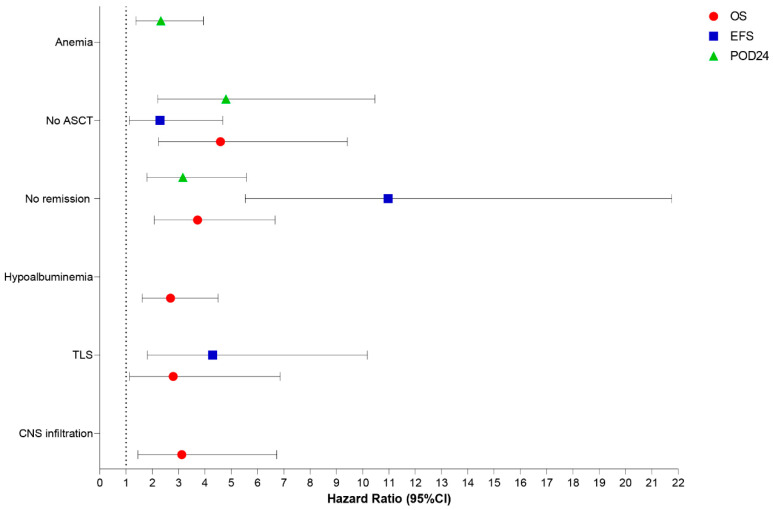
Independent prognostic factors for OS (red), EFS (blue), and POD-24 (green) in 165 MCL patients identified in multivariate analysis. ASCT: autologous stem cell transplantation; TLS: tumor lysis syndrome; CNS: central nervous system.

**Table 1 cancers-15-04759-t001:** Clinical-epidemiological characteristics of 165-MCL patients.

Characteristic	N = 165; % (N)
Male gender	73.9% (122/165)
Age (median, range)	65 years (38–89 years)
Elderly (age ≥ 60 years)	66.7% (110/165)
Caucasian ethnicity	72.1% (119/165)
MCL clinical–biological subtype	cnMCL: 90.9% (150/165) nnlMCL: 9.1% (15/165)
PB involvement	47.3% (78/165)
BM infiltration	76.4% (120/157)—N miss = 8
CNS infiltration	6.1% (10/165)
Gastrintestinal involvement	22.4% (37/165)
Waldeyer’s ring involvement	20% (33/165)
Orbit infiltration	3.6% (6/165)
Skin/SCT involvement	4.8% (8/165)
Splenomegaly	56.4% (93/165)
Bulky disease ≥ 7 cm	32.7% (54/165)
Ann Arbor clinical stage	I/II—4.2% (7/165); III/IV—95.8% (158/165)
ECOG	<2–67.9% (112/165); ≥2–32.1% (53/165)
B-symptoms	64.8% (107/165)
TLS	6.1% (10/165)
≥2 comorbidities	49.1% (81/165)
MIPI score—median (range)	6.4 points (5.4–13.7 points)
MIPI score—risk-categories	Low-risk (<5.7) = 13.3% (22/165) Intermediate-risk (5.7–6.1) = 22.4% (37/165) High-risk (≥6.2) = 64.2% (106/165)

MCL: mantle cell lymphoma; cnMCL: classic nodal mantle cell lymphoma; nnlMCL: non-nodal leukemic MCL; PB: peripheral blood; BM: bone marrow; CNS: central nervous system; SCT: subcutaneous tissue; ECOG: *Eastern Cooperative Oncology Group* scale; TLS: tumor lysis syndrome; MIPI: *Mantle Cell Lymphoma International Prognostic Index*.

**Table 2 cancers-15-04759-t002:** Laboratory characterization of 165-MCL patients.

Laboratory Parameter	Median (Range)
Hemoglobin (g/L)	114 (3.4–16.9)
WBC (×10^9^ cels/L)	8.72 (3.80–750.04)
Neutrophils (×10^9^ cels/L)	3.75 (0–52.1)
Lymphocytes (×10^9^ cels/L)	2.51 (0.05–676.7)
Monocytes (×10^9^ cels/L)	0.6 (0–15.0)
N/L ratio	1.34 (<0.01–50)
L/M ratio	4.26 (<0.01–98)
Platelets count (×10^9^ cels/L)	160 (2–552)
LDH (U/L)	357 (142–10,870)
Patient LDH/control LDH ratio	1.07 (0.31–22.6)
β2- microglobulin (mg/dL)	3.80 (1.07–46.4)
Albumin (g/dL)	4.0 (1.3–5.4)
Globulin (g/dL)	2.5 (0.5–8.8)
**Laboratory parameter**	**Characterization—% (N)**
Anemia (Hb < 120 g/L)	57% (94/165)
Leukocytosis (WBC > 11.0 × 10^9^/L)	37.6% (63/165)
Neutrophilia (>7.0 × 10^9^/L)	17% (28/165)
Lymphocytosis (>3.0 × 10^9^/L)	43% (71/165)
Monocytosis (>1.0 × 10^9^/L)	25.5% (42/165)
LDH ≥ UVN	55.8% (92/165)
β2-microglobulin ≥ UVN	67.1% (88/131)—N miss = 34
Hypoalbuminemia (<3.5 g/dL)	72.9% (113/155)—N miss = 10
Hypogamaglobulinemia (<1.5 g/dL)	2.6% (4/150)—N miss = 15
Paraproteinemia (M-spike)	5.5% (9/154)—N miss = 11
HIV positive serology	0.6% (1/158)—N miss = 7
HbsAg positive antigen	0% (0/155)—N miss = 10
HCV positive serology	1.3% (2/155)—N miss = 10

WBC: white blood cell count; N/L: neutrophil/lymphocyte ratio; L/M: lymphocyte/monocyte ratio; LDH: lactic dehydrogenase.

**Table 3 cancers-15-04759-t003:** Histopathological and immunohistochemical aspects of 165-MCL patients.

Pathological/Immunohistochemical Finding	% (N)
Diagnostic biopsy site	Lymph node—50.9% (84/165) BM—25.5% (42/165) Extra-lymphoid tissues—23.6% (39/165)
Blastoid morphology	17.6% (29/165)
High-index Ki-67 (≥30%)	56.5% (65/115)—N miss = 50
CD5^+^	91.5% (150/164)—N miss = 1
CD23^+^	6.8% (9/132)—N miss = 33
CD200^−^	93.2% (69/74)—N miss = 91
Cyclin-D1^+^	95.1% (157/165)

BM: bone marrow; CD: cluster designation.

**Table 4 cancers-15-04759-t004:** Therapeutic management adopted in 165-Brazilian MCL patients included in the study.

Characteristic	% (N)
Anti-cancer therapy	92.1% (152/165)
Up-front therapy	
-(R)-CHOP-like induction	46% (70/152)
-(R)-HDAC induction regimens	43.4% (66/152)
* (R)-HyperCVAD	37.8% (25/66)
* (R)-CHOP/(R)-DHAP	62.2% (41/66)
-Clinical trials (VR-CAP, BRi, BRa)	5.3% (8/152)
-Chorambucil monotherapy (palliative)	4.6% (7/152)
-Isolated IF-RT	0.7% (1/152)
-WW approach	3.0% (5/165)
-Without any therapy (by early-deaths)	4.9% (8/165)
≥2 lines of treatment	51.9% (79/152)
ASCT consolidation	32.8% (50/152)
-Up-front ASCT	82% (41/50)
-Not in 1st. CR/PR	18% (9/50)
Allogeneic SCT	3% (5/165)
Rituximab during induction therapy	21.2% (32/151)
Adjuvant IF-RT	7.3% (11/151) *
Cytoreduction	41.4% (63/152)
Intrathecal CNS prophylaxis	22.3% (34/152)

* not included 1 patient underwent isolated IF-RT; R-CHOP (rituximab, cyclophosphamide, doxorubicin, vincristine, and prednisone); R-HyperCVAD (course A: similar to R-CHOP with cyclophosphamide fractionation dose; course B: rituximab, high-dose methotrexate plus high-dose cytarabine); R-DHAP (rituximab, dexamethasone, high-dose cytarabine plus cisplatin); VR-CAP (bortezomib, rituximab, cyclophosphamide, adryamicin, prednisone); BRi (bendamustine, rituximab plus ibrutinib); BRa (bendamustine, rituximab plus acalabrutinib); WW: *watchful and waiting approach*; ASCT: autologous stem cell transplantation; CR: complete remission; PR: partial response; SCT: stem cell transplantation; IF-RT: involved-field radiotherapy; CNS: central nervous system.

**Table 5 cancers-15-04759-t005:** Responses and clinical outcomes in the main cohorts of MCL patients.

Clinical Study	N	Up-Front Therapy	Responses	Survival
Bosch et al., 1998 [34]	59	Anthracyclines without ASCT	ORR: 65% CR: 19%	Median OS: 29 months
Sarkozy et al., 2014 [35]	125	R-CHOP and R-CHOP/R-DHAP (36% underwent ASCT)	ORR: 92% CR: 69%	Median OS: 73 months
Abrahamsson et al., 2011 [36]	785	Mainly R-CHOP	ORR: 71% CR: 47%	3-year OS: 62%
Romaguera et al., 2010 [37]	97	R-HyperCVAD without ASCT	ORR: 97% CR: 87%	Median OS: 127 months
Hermine et al., 2016 [25]	466	R-CHOP plus ASCT (*n*-234) R-CHOP/R-DHAP plus ASCT (*n* = 232)	ORR:97%/CR:61% ORR:98%/CR:63%	Median OS: NR Median OS: 116 months
Flinn et al., 2014 [38]	94	BR (*n* = 37) R-CHOP (*n* = 37)	ORR:97%/CR:40% ORR:91%/CR:30%	Median OS NR in both arms
HC-FMUSP from 2010 to 2022 (This study)	165	R-HDAC (*n* = 66) R-CHOP (*n* = 70) (32.8% underwent ASCT)	ORR: 85.9% ORR: 65.7%	2-year OS: 73% 2-year OS: 58%

ASCT: autologous stem cell transplant; ORR: overall response rate; CR: complete response; OS: overall survival; R-CHOP: rituximab, cyclophosphamide, doxorubicin, vincristine, and prednisone; R-DHAP: rituximab, dexamethasone, cytarabine, cisplatin; R-HyperCVAD: rituximab, hyperfactioned cyclophosphamide, vincristine, adryamicin, and dexamethasone alternating with HDAC and HD-methotrexate; BR: bendamustine plus rituximab.

**Table 6 cancers-15-04759-t006:** Main adverse prognostic factors associated with decreased OS (multivariate analysis) in different MCL cohorts.

Clinical Study	Study Design	N	OS Prognostic Factors
Bosch et al., 1998 [34]	Retrospective	59	Poor performance status, splenomegaly, mitosis index > 2.5
Samaha et al., 1998 [39]	Retrospective	121	Age ≥ 60 years, Hb < 120 g/L, poor performance status, peripheral blood involvement
Sarkozy et al., 2014 [35]	Retrospective	125	MIPI high-risk, complex karyotype, blastoid morphology
Andersen et al., 2002 [40]	Retrospective	105	Age ≥ 60 years, Hb < 120 g/L, splenomegaly
Hoster et al., 2016 [10]	Retrospective	508	Ki-67 > 30%; blastoid morphology, MIPI high-risk
Abrahamsson et al., 2011 [36]	Retrospective	785	Age ≥ 60 years, poor performance status, B-symptoms
HC-FMUSP from 2010 to 2022 (This study)	Retrospective	165	CNS infiltration, TLS, albumin < 3.5 g/dL, up-front ASCT omission, absence of response after 1st. line

OS: overall survival; Hb: hemoglobin; MIPI: Mantle Cell Lymphoma International Prognostic Index; CNS: central nervous system; TLS: tumor lysis syndrome; ASCT: autologous stem cell transplantation.

## Data Availability

All data generated and analyzed during this study were included in this published article. The raw data for this study are in the possession of the correspondence author and may be fully available in the event of a request to the correspondence author via e-mail.

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
