# Peer review of "Up-Front ASCT Overcomes the Survival Benefit Provided by HDAC-Based Induction Regimens in Mantle Cell Lymphoma: Data from a Real-Life and Long-Term Cohort"

_cancers, 2023, doi:10.3390/cancers15194759_

Round 1
Reviewer 1 Report
Mantle cell lymphoma remains incurable and the standard of care for young and fit patients involves induction immunochemotherapy followed by up-front autologous stem cell transplantation (ASCT). As presented by the outhors, the role of high-doses of cytarabine remains controversial.
In this paper the outhors compare, in a small setting of patients , and in small period ( just two years) compare responses between different therapeutic strategies, focusing on the impact of HDAC-based regimens on outcomes in ASCT-eligible patients.
Good results and good methods are presented.
Reviewer 2 Report
In their manuscript „Up-front ASCT overcomes the survival benefit provided by HDAC-based induction regimens in mantle cell lymphoma: data from a real-life and long-term cohort“ Covas Lage et al discuss the OS and EFS in a brazilian cohort between 2010 and 2022. Covas Lage et al analyzed a cohort comprised of 165 ASCT-eligible patients diagnosed with histological proven MCL. This study claims to be the largest real-world Latin American study ever reported, which confirms the importance of up-front consolidation with ASCT in the MCL management, regardless of the induction immunochemotherapy regimen used
Major:
In comparison to Hermine O, Dreyling M et al. Addition of high-dose cytarabine to immunochemotherapy before autologous stem-cell transplantation in patients aged 65 years or younger with mantle cell lymphoma (MCL Younger): a randomized, open-label, phase 3 trial of the European Mantle Cell Lymphoma Network. Lancet. 2016; 388(10044): 565-575 the smaller number of cases reduces the option for a meaningful interpretation of the demonstrated data. The long-term follow-up data of Hermine et al (DOI: 10.1200/JCO.22.01780 Journal of Clinical Oncology 41, no. 3 (January 20, 2023) 479-484) should be presented and discussed.
Limitations exist when retrospective studies, as the one submitted had a cohort that was submitted to heterogeneous therapeutic modalities. Therefore randomisation (R-CHOP vs R-HDAC-like) shows some inaccuracy as two protocolls ((R)-HyperCVAD and (R)-CHOP/(R)-DHAP) have been used for the focus group. Therefore statistical power is unnecessarily reduced.
Covas Lage et al have included first-line treatment as well as advanced treatment line (>2 treatment lines before). Preferrably only first line treatment or second line treatment should be included.
As stated also trial study protocols such as VR-CAP, BRi, BRa have been included to a total of 5.3%.
Minor:
As Rituximab maintenance regimen is not permitted/approved by Brazilian administration/ health officials, comparison to European or North-American studies show significant reduction in OS and PFS in Brazilian vs European/ North-American focus groups.
Figure 3 A and B needs reconfiguration as layout prevents proper readability.
Spelling mistakes have been made in Table 2 (e.g. Hipoalbuminemia).
Minor adjustements need to be made
Round 2
Reviewer 2 Report
Corrections have been applied by the authors. All mentioned suggestions for improvements have been implemented.
Minor corrections can be applied to improve the quality of english language in the already corrected discussion part.